# Attachment Security and Suicide Ideation and Behaviour: The Mediating Role of Reflective Functioning

**DOI:** 10.3390/ijerph18063090

**Published:** 2021-03-17

**Authors:** Jessica Green, Katherine Berry, Adam Danquah, Daniel Pratt

**Affiliations:** 1Division of Psychology and Mental Health, School of Health Sciences, Manchester Academic Health Science Centre, University of Manchester, Manchester M13 9PL, UK; jessica.green19@nhs.net (J.G.); katherine.berry@manchester.ac.uk (K.B.); adam.danquah@manchester.ac.uk (A.D.); 2Greater Manchester Mental Health NHS Foundation Trust, Manchester M25 3BL, UK

**Keywords:** adult attachment, suicide, reflective functioning, mentalisation, mediation

## Abstract

Background: To understand why attachment difficulties predispose individuals to suicidal thinking (suicide ideation) and behaviour, a leading cause of death, we need to explore the role of pertinent psychological mechanisms. Attachment processes are closely linked to the development of mentalisation capabilities, or reflective functioning; the ability to understand and interpret self and other behaviour as an expression of mental states. Interventions designed to improve mentalisation have been associated with a reduction in suicidal behaviour, yet reflective functioning has not been directly investigated in relation to suicidal ideation and behaviour. Aim: We aim to further verify the link between adult attachment security and suicidal ideation and examine whether deficits in reflective functioning mediate this relationship. Methods: Sixty-seven participants who experienced suicidal ideation within the past 12 months completed self-report measures of adult attachment, current suicidal ideation, reflective functioning, depressive symptomology and hopelessness. Partial correlations, mediation analyses and group comparisons were conducted to explore relationships between these factors. Results: Findings did not support a mediational role for reflective functioning in the relationship between attachment security and suicidal ideation. A direct relationship was established between attachment avoidance and suicidal ideation, after controlling for age, gender and depressive symptoms. However, participants with a history of attempted suicide were higher in anxious attachment compared to participants with no such history. Conclusions: This study shows that the attachment dimensions of attachment anxiety and avoidance may play differential roles in increasing risk for suicidal attempts versus ideation. This has important implications for tailoring interventions. Those aimed at reducing suicide attempts should focus on reducing attachment anxiety by helping people develop skills in emotional regulation. Interventions aimed at reducing suicidal ideation should focus on reducing attachment avoidance by helping people develop closer relationships with significant others. However, longitudinal and experimental designs are required to verify causality.

## 1. Introduction

Suicidal behaviour is a major global health concern. Each year, approximately 800,000 people die by suicide, making it the leading cause of death worldwide among 15–29 year olds [1]. In 2019, 5691 suicides were registered in England and Wales, with the highest proportion in males aged 40–44 years [2]. As suicide is the result of a person deciding to take action to end their own life, it is perhaps the cause of death most directly affected by psychological factors. Therefore, improving our understanding of the psychological processes that underpin an individual’s thoughts about suicide (suicidal ideation) and related behaviours, including suicide attempts, is essential for early identification of risk and development of effective psychotherapeutic interventions.

Although the causes of suicide are not fully understood, it is generally accepted that suicidal ideation and behaviours result from the complex interplay of many factors [3]. Psychological explanations have been developed to improve our understanding of suicide. Initial explanations emphasised the central role of social connectedness, with a lack of social integration proposed to increase the likelihood of suicide [4]. More contemporary models propose a diathesis–stress framework with a central focus upon cognitive factors [5,6,7,8,9,10]. Most recent suicide theories have been set the challenge of differentiating between individuals who engage in suicidal ideation alone from those whom also engage in suicidal behaviours. The distinction proposed by this ideation-to-action framework has proven to be especially significant given that most people who think about suicide do not engage in suicide behaviour [11].

Importantly, the Interpersonal-Psychological Theory of Suicidal Behaviour (IPT) [12,13] presented the first desire–capability framework that explained how a desire for suicide could emerge from a sense of social alienation (low belongingness) and burdensomeness, but this desire would only be realised (progression from ideation to behaviour) in the presence of a capability to act on such desires. A meta-analysis of 122 published and unpublished studies [14] examined the decade of research that has subsequently investigated the relationship between IPT constructs and suicidal ideation and behaviours. Overall, the meta-analysis offered support for IPT, admittedly with weak-to-moderate positive associations reported for thwarted belonginess (r = 0.37 and r = 0.11) and perceived burdensomeness (r = 0.48 and r = 0.25) with suicidal ideation and suicide attempt history, respectively. Capability for suicide was also significantly associated with suicide ideation and attempts although these effects were weak (rs = 0.09–0.10). Further, in addition to these univariate effects, the interaction of thwarted belongingness and perceived burdensomeness was significantly, but weakly, correlated with suicide ideation (rs = 0.12–0.14). Similarly, the three-way interaction of all IPT constructs was significantly associated with a greater number of suicide attempts, although this interaction effect was also weak (rs = 0.06–0.11). The authors of the review concluded that these findings are largely consistent with the IPT hypotheses, albeit with effect sizes no better that those reported for the “many traditional and often-studied risk factors (e.g., suicide attempt history, demographic variables, psychiatric diagnoses, social factors)” (p. 1332). The dominance of Joiner’s IPT [12] within the suicide literature is not without criticism though. For example, Hjelmeland and Knizek [15] recently highlighted that the Chu et al. (2017) meta-analysis was only able to offer limited support for the IPT and suggested the conclusions to be drawn from this review were “clearly strained”. Indeed, a previous systematic review of studies investigating IPT also suggested that this theory of suicide “may not be as clearly defined nor supported as initially thought” (p.40) [16].

Current understandings of suicide, therefore, seek to explain not only the initial development of suicidal ideation but also how ideators can be distinguished from suicide attempters [7,11]. However, theorists have tended to overlook developmental perspectives, for example, how attachment security may contribute to disruptions in relationships and lead to the emergence of suicide ideation and/or suicide behaviour.

Attachment theory is a useful framework for understanding how early experiences of caregiving shape future feelings of security and behaviour in interpersonal relationships. Bowlby [17] proposed that children who receive responsive and consistent care develop secure mental representations (or internal working models) of others as available and supportive, and themselves as loved and capable. In contrast, infants who experience care that is insensitive, inconsistent or rejecting will learn to view others as unavailable or unpredictable, and themselves as unlovable. Through repeated interactions, these representations become entrenched, and guide how infants activate their attachment system in times of danger or distress, and their expectations of future interpersonal exchanges [18]. Bowlby’s work was extended by Ainsworth and colleagues [19,20], who, in addition to secure attachment, identified two distinct styles of insecure attachment: anxious and avoidant. Whereas secure infants activate their attachment system appropriately upon separation and quickly return to baseline, anxious children maximise their distress signals and are difficult to soothe when reunited. In contrast, avoidant infants display minimal distress and shift their attention away from their mother when she returns.

Research across the lifespan indicates that these internal representations and attachment styles remain moderately stable into adulthood [21]. Adult attachment theory assumes that patterns developed in the context of parent–child relationships translate into similar styles of relating in the context of romantic relationships [22]. Extending this, Bartholomew and Horowitz [23] proposed a four-factor model that parallels the categories observed in infants. Depending on whether adults view themselves and others as positive or negative, they can be categorised as secure, preoccupied, dismissing or fearful. However, there has been a move towards conceptualising attachment differences on dimensions rather than categories [24]. Individuals high in attachment avoidance are uncomfortable with closeness in relationships and overvalue independence, whereas those high in attachment anxiety strongly desire close relationships yet have an intense fear of abandonment. Individuals low in both attachment anxiety and avoidance are securely attached; they feel close to significant others and can rely on them in times of need.

Adams [25] put forward a developmental model that conceptualised suicide as an extreme attachment behaviour, signalling distress and anger towards an inconsistent or unavailable attachment figure. This model proposed that when experiencing current threat or distress, individuals with trait vulnerabilities of anxious or avoidant attachment are unable to draw on resources from interpersonal relationships as efficiently as their securely attached peers, and instead resort to suicidal thinking or behaviour as the crisis escalates. Furthermore, insecurely attached individuals will have greater sensitivity to interpersonal threat such as loss, disappointment and rejection, which will lead to more frequent activation of their attachment system. For avoidant individuals, suicide may be the eventual outcome of a deactivated attachment system, where they have become socially isolated because of avoiding close relationships and eventually rejecting life itself [26]. Alternatively, anxiously attached individuals, who crave closeness but fear abandonment, may resort to suicidal gestures and behaviours to elicit care and support from others in the absence of more adaptive strategies.

Empirical research has reliably demonstrated that attachment insecurity is a general risk factor for many psychological difficulties [27,28], and there is a growing body of literature which supports a relationship between insecure attachment and suicide-related outcomes [29]. This has been evidenced in research with adolescents [30,31,32], adults [33,34], and for suicidal ideation [35,36] and attempts [34,37,38,39]. A review of sixteen studies examining adult attachment and suicidal ideation and attempts found that predominately anxious styles were associated with an increased suicide risk and concluded that suicidality is the result of an interaction between long-lasting insecure attachment patterns and current symptoms of various psychopathologies [26].

However, to better understand why attachment difficulties predispose individuals to suicidal ideation and behaviour, we need to examine psychological mechanisms that are theoretically proposed to mediate this association. Exploratory research has begun to investigate psychological constructs that may bridge this gap, including interpersonal problems [40], self-criticism and dependency [41], loneliness [42] and feelings of entrapment [43,44]. Yet this is an emerging body of literature with minimal consensus or underpinning theoretical models. Adams’ model [25] outlines a number of psychological factors that could intervene between attachment security and later suicidal behaviour, including personal vulnerability and resilience factors, and skill deficits that are a potential consequence of adverse parenting experiences. However, since its development, further theoretical advances have been made that may have been overlooked in the original conceptualisation.

One psychological construct that is intimately linked with attachment is mentalisation; operationalised by Fonagy and colleagues as ‘reflective functioning’ [45,46]. Mentalisation, or reflective functioning (used synonymously), refers to the human capacity to understand and interpret one’s own behaviour, and the behaviour of others, as expressions of mental states such as thoughts, feelings, beliefs and desires [46]. Having the ability to form relatively accurate models of the mind, whilst acknowledging the opaqueness of mental states, helps individuals understand and anticipate one another’s actions [45,47]. Fonagy and colleagues further outline two subtypes of mentalisation impairment: hypomentalisation, the extreme difficulty developing complex models of the mind of oneself and others, and hypermentalisation, the opposite tendency to develop very complex models that have little or no correspondence to observable evidence. Genuine mentalisation is a vital skill that allows people to successfully navigate their social world and regulate their affect [46], and both hypo- and hypermentalising have been implicated in a wide range of psychological disorders [48].

Refinement of this skill and its robustness in highly distressing emotional interactions is influenced by early attachment experiences [46]. To develop mentalising skills, children need to experience sensitive and attuned care from somebody who has their mind in mind (mind-mindedness [49]). This provides the context for infants to become sensitised to their own inner self-states, and the mental states of others. As reflective functioning develops, others’ behaviour becomes more predictable and meaningful, which enables individuals to respond flexibly and adaptively to interpersonal interactions [47]. An abusive or neglectful early environment, which often underpins insecure attachment, can disrupt the acquisition of important mentalising skills [46]. Furthermore, adults with insecure attachment continue to show fluctuations in their capacity to mentalise, especially when their attachment system is aroused [50].

Contemporary theories have largely focused on attachment disruptions and mentalisation deficits in relation to borderline personality disorder (BPD). Fonagy and Luton’s [50] mentalisation-based model postulates that distal (attachment disruptions) and proximal risk factors (stress and arousal) interact to lower a person’s threshold for activation of their attachment system, and subsequent deactivation of their mentalising capabilities. When mentalisation skills are ‘switched off’, individuals become vulnerable to the core features of BPD; affect dysregulation, poor impulse control, dysfunctional relationships, dissociation and feelings of inner pain and emptiness [45,50]. Many of these core features have been established as key risk factors for suicide [51,52,53] and there are similarities between Fonagy and colleagues’ conceptualisation of BPD and Adams’ [25] developmental model of suicide. In the context of Adams’ model, mentalisation impairments would be understood as a consequence of insecure attachment that increases vulnerability to suicidal thinking and behaviour when coupled with acute stress or interpersonal difficulties. Moreover, a study of BPD patients found that those who received mentalisation-based treatment to improve their reflective functioning made significantly fewer suicide attempts over an 8 year follow-up period compared to those who received treatment as usual [54]. This suggests that impaired reflective functioning may be at least partly responsible for increased suicidal behaviours, and more importantly, that suicide risk could be reduced through effective psychological intervention.

In a review of reflective functioning, Katznelson [48] found mentalisation impairments to associate with various forms of psychopathology. However, there is a shortage of research that has explicitly investigated the link between reflective functioning and suicidal thoughts or behaviour. Studies that have attempted to explore this association have assessed conceptually related constructs as a proxy for reflective functioning (e.g., Alexithymia, Theory of Mind) which limits the validity of their conclusions [55]. Research efforts may have been hampered by the fact that until recently assessment of reflective functioning relied on administering and rating the Adult Attachment Interview (AAI) [56], an expensive and time-consuming assessment process [48]. However, the recent development of the Reflective Functioning Questionnaire (RFQ) [57], a brief self-report measure of mentalising, should enable more valid and convenient measurement of this complex psychological variable.

Based on the previous theoretical arguments and empirical findings, the present study aimed to further verify the relationship between adult attachment security and suicidal ideation and examine the potential mediating role of reflective functioning. Suicidal ideation was chosen as the primary outcome variable as it is more prevalent in the general population [58] and a key risk factor for eventual suicide [59]. Approximately one-third of individuals who experience suicidal thoughts go on to attempt suicide, and 60% of transitions from ideation to attempt occur within the first year of ideation onset [58]. Furthermore, the severity of ideation is associated with a higher likelihood of future suicidal behaviour [60]. Suicidal thinking also causes significant distress in its own right; experiencing enduring suicidal ideation has been shown to increase the likelihood of impaired psychosocial and mental functioning in young adulthood [61]. Therefore, it is imperative that we better understand the psychological mechanism that underpin this relevant precursor in order to prevent impaired psychosocial functioning and subsequent suicidal behaviour.

In addition to the primary mediation analyses, and in accord with the suicide ideation-to-action framework [11], additional analyses were also carried out to investigate individual differences between participants based on their self-reported histories of suicide attempts. Our specific hypotheses were:

**Hypothesis** **1** **(H1).***Anxious and avoidant attachment will be positively associated, with a moderate to large effect, with suicidal ideation; participants scoring higher on self-report measures of these attachment dimensions will also score higher on a measure of recent suicidal ideation, after adjustments have been made for key sociodemographic and psychological variables*.

**Hypothesis** **2** **(H2).***The relationship between attachment security and suicidal ideation will be significantly, with a moderate effect, mediated by deficits in reflective functioning*.

**Hypothesis** **3** **(H3).***Participants with a self-reported history of attempted suicide will score significantly higher, with a moderate effect, on measures of attachment security and reflective functioning*.

## 2. Materials and Methods

The current study formed part of The CLoseness to Others and Suicidal Experiences (CLOSE) project, a collaborative research project at the University of Manchester. As such, not all measures administered to participants are reported in the current study. The research protocol described here was reviewed and approved by the North West—Greater Manchester West NHS Research Ethics Committee (Ref: 17/NW/0194) and the Health Research Authority.

### 2.1. Sampling and Procedure

Participants identified through the National Health Service (NHS) were recruited from two NHS trusts in the North West of England. Recruitment efforts were targeted at community outpatient and inpatient mental health services, as individuals accessing these services are more likely to have an increased risk for suicidal ideation. NHS clinicians shared information about the project with eligible patients and sought consent for a researcher to approach them in person or via telephone. Patients could also self-refer to the project using contact details provided by their clinician, or from posters and leaflets displayed at approved NHS sites. Study advertisements were also displayed in third-sector voluntary organisations and public places (e.g., University of Manchester campus). Participants who self-referred to this study were required to provide contact details of a responsible clinician so any risk concerns arising from participation could be shared if required.

Once identified as eligible, a researcher met with the potential participant to provide more information and answer any questions. Following consent, participants were given the option to complete the questionnaires independently or with assistance from the researcher. Upon completion, participants were provided with a debriefing sheet containing crisis advise and contact information and given the opportunity to discuss and reflect on the experience. Any risk concerns arising during participation were handed over to a member of the participant’s care team.

### 2.2. Inclusion Criteria

The primary inclusion criterion was self-reported suicidal ideation within the past year. This timeframe is commonly adopted in research examining suicidal ideation [62] to capture a sample where suicidal thinking is a recent or ongoing experience, balanced against what is pragmatic for recruitment purposes. A positive response to the screening question ‘have you had any thoughts of killing yourself in the past 12 months’ was used to confirm eligibility. Participants were also asked ‘approximately, when was the last time you had any thoughts of killing yourself?’ to gather information on the recency of their suicidal ideation. Further inclusion criteria included being 18 years or above, having sufficient English language proficiency and having capacity to provide informed consent as established through clinical observations at the time of interview. Exclusion criteria included a primary organic mental disorder (e.g., traumatic brain injury, dementia) and judged by the researcher to be intoxicated at the time of interview. No financial or alternative incentive was offered for taking part.

### 2.3. Measures

In total, seven measures and a sociodemographic questionnaire, developed for this study, were administrated to participants. The administration of the measures was counterbalanced to reduce order effects. The measures included in the current study are described below.

The Beck Scale for Suicidal Ideation (BSSI) [63] is a 21-item self-report measure that assesses suicidal thinking and planning over the past week. Each item has three response options (e.g., ‘I have no wish to die’, ‘I have a weak wish to die’, or ‘I have a moderate to strong wish to die’) which are scored from 0 to 2. Typically, the first five items are used as screening questions in non-clinical samples. However, in the current sample, participants were asked to respond to all items. Responses to items 1–19 were summed to provide a total ideation score ranging from 0 to 38, with a higher score indicating more severe suicidal thinking. Items 20 and 21 indicate whether the respondent has a history of suicide attempt(s), and were not included in the total ideation score. The BSSI has demonstrated excellent internal consistency in a clinical sample of mood disorder patients (α = 0.97) [64].

The Revised Experiences in Close Relationships Scale (ECR-R) [65] is a 36-item self-report measure of adult attachment security. Respondents rate on a 7-point Likert scale their agreement with statements about how they generally experience close relationships. The questionnaire includes two subscales that assess dimensions of attachment-related security; avoidance and anxiety (18 items each). Items that make up the anxiety subscale (e.g., ‘I worry about being abandoned’) measure the extent to which individuals fear abandonment, have a negative self-view and are highly preoccupied with romantic partners. Alternatively, items that measure attachment avoidance (e.g., ‘I prefer not to show a partner how I feel deep down’) measure the degree to which individuals avoid intimacy, view others negatively and do not seek support when required. High scores on either dimension indicate greater attachment insecurity with mean scores ranging from 1 to 7. The ECR-R subscales have been found to have high internal consistency with Cronbach’s α coefficients of near or above 0.90 [66], and highly stable test–retest reliability [67].

The Reflective Functioning Questionnaire (RFQ) [57] is a brief screening measure of mentalisation capabilities, made up of eight items that participants rate on a 7-point Likert scale ranging from ‘strongly disagree’ to ‘strongly agree’. The scoring procedure yields two subscales that measure hyper- and hypomentalising. Six of the eight items are included on both scales but are scored differently to capture the different failures in mentalisation. The Certainty about Mental States (RFQ-C) subscale includes items such as ‘People’s thoughts are a mystery to me’ which are reverse scored to capture extreme levels of certainty. Very low levels of agreement with RFQ-C items reflect distorted, projective mentalising (hypermentalising), while some agreement reflects adaptive levels of certainty about mental states. The Uncertainty about Mental States (RFQ-U) subscale includes items such as ‘Sometimes I do things without really knowing why”. High levels of agreement with RFQ-U items reflect an inability to consider complex models of one’s own mind or others, or hypomentalising, whereas lower scores indicate an awareness of the opaqueness of one’s own mental states and those of others, typical of genuine mentalising. The RFQ-C and RFQ-U subscales have shown acceptable internal consistency (Cronbach’s α = 0.73 and 0.78, respectively) in a clinical sample and have been found to significantly relate in theoretically predicted ways with related constructs of empathy, mindfulness and perspective-taking [57].

The Beck Hopelessness Scale (BHS) [68] is a 20-item self-report inventory that assesses three aspects of hopelessness—negative beliefs about the future, loss of motivation and expectation. Participants rate pessimistic (e.g., ‘my future seems dark to me’) and optimistic (e.g., ‘I look forward to the future with hope and enthusiasm’) items as either true or false in relation to how they have felt in the past week. Items are scored 0 (false) or 1 (true), and positive items are reverse scored. Items are summed to produce a total score ranging from 0 to 20, with a higher score indicative of a greater severity of hopelessness. The scale has shown excellent internal reliability (α = 0.93) in clinical samples [68], and adequate convergent, discriminant and predictive validity in a meta-analysis [69].

The Patient Health Questionnaire- 9 (PHQ-9) [70] is a brief 9-item self-report measure that assesses depression symptom severity. Respondents are instructed to rate how often they have experienced common symptoms of depression (e.g., feeling down, depressed or hopeless) over the past two weeks, on a 4-point Likert scale from 0 (not at all) to 3 (nearly every day). The items are summed to give a total score which can range from 0 to 27, with a higher score indicative of greater depression severity The scale has demonstrated good internality reliability (α = 0.86–0.89), excellent test–retest reliability and predictive validity for major depression [70].

### 2.4. Statistical Analysis

Patterns of missing data were explored using the Missing Values Analysis function in IBM SPSS Statistics (v. 23). No missing data were found for the PHQ-9, and only 1 data point was missing for the RFQ. Little’s chi-square statistic was non-significant for the other scales, indicating data were missing completely at random (MCAR). Therefore, Expectation-Maximisation (EM) method, a method of single imputation, was used to estimate and replace small amounts of missing data (<20% per participant, per scale) to retain the maximum number of participants for analysis.

Bias checks were conducted to assess for outliers and non-normal distribution of data. The Kolmogorov–Smirnov test was carried on all total scales and subscales to assess for normal distribution. The ECR-R subscale scores for anxious (D (64) = 0.78, *p* = 0.20) and avoidant (D (64) = 0.76, *p* = 0.20) attachment security did not deviate significantly from a normal distribution; however, scores for depression, hopelessness, suicidal ideation, and reflective functioning were all significantly non-normal. Total scales and subscales had acceptable-to-excellent internal consistency (α = 0.78–0.94), with Cronbach alpha statistics being comparable to those found in previous research studies.

Descriptive characteristics were calculated for each variable, in addition to partial correlations controlling for age and gender. Where assumptions of univariate normality were violated, 1000 bootstrapped samples were drawn, and bias-corrected 95% bootstrap confidence intervals (CIs) reported. Bootstrapping is a nonparametric resampling procedure and is recommended as an alternative to transforming data or other non-parametric tests when parametric assumptions are violated [71].

Between-group comparisons based on participant’s suicide attempt history were also carried out for measures of attachment security and reflective functioning. The Kolmogorov–Smirnov test was used to assess within-group data distribution. Where normality was violated the Kruskal–Wallis test was ran as the non-parametric alternative, as the F statistic reported in an ANOVA cannot be bootstrapped [71]. Due to the small n and unequal group sizes, it was not appropriate to carry out further analyses with group allocation as the dependent variable.

To assess hypothesis 2, that the relationship between attachment security and suicidal ideation is mediated by deficits in reflective functioning, simple mediation models were applied. Mediation analyses were performed using Hayes’ PROCESS (2.16.3) model 4 for SPSS [72]. Gender, age and self-reported depressive symptoms were included as covariates in the mediation model to adjust for the potential effects of these factors on both reflective functioning and suicidal ideation. In all cases, 1000 bootstrapped samples were used to generate a sampling distribution and a 95% confidence interval for the indirect effect; statistical significance of the indirect effect is determined by the absence of zero from the confidence interval [71].

## 3. Results

In total, 67 participants completed the questionnaire measures. Two participants were excluded as they reported last experiencing suicidal thoughts over 1 year ago. Therefore, 65 participants were included in the final sample. Full sociodemographic characteristics of the sample are presented in Table 1. Participants were aged between 18 and 63 years, with a mean age of 32.15 (Standard Deviation = 12.45) years. The sample were predominantly White British, female and currently single (67.7%).

Most participants self-reported at least one psychiatric diagnosis, with mood disorders (e.g., depression, bipolar), anxiety disorders (e.g., anxiety, post-traumatic stress disorder, social anxiety) and personality disorders (e.g., borderline/emotionally unstable personality type) most commonly reported. Twenty-four participants also reported having a disability (36.9%), which included physical disabilities (e.g., chronic health conditions, mobility impairments), learning difficulties or disabilities (e.g., mild learning disability, dyslexia and dyspraxia), autistic spectrum disorders and mental health difficulties (when the participant considered this to be a disability).

Descriptive statistics for all questionnaire measures and the results of normality and reliability tests are reported in Table 2. Participants self-reported on a single item measure the recency of their suicidal ideation. Forty-five participants reported experiencing suicidal thoughts within the past month (69.2%). Three quarters of the sample also reported a lifetime history of attempted suicide (73.8%), as measured by item 20 on the BSSI. Of the 48 participants who reported a past suicide attempt, 29 (60.4%) reported having attempted suicide on multiple occasions.

Partial correlations were carried out between all study variables using Pearson product–moment correlation coefficients, controlling for age and gender (Table 3). To account for non-normal distributions of data, bias-corrected and accelerated (BCa) bootstrap 95% confidence intervals are reported in square brackets. Moderate to strong positive correlations [73] were found between measures of recent suicidal ideation, hopelessness and depression. Weaker associations with anxious attachment were found for current depression (r = 0.26), hopelessness (r = 0.27) and suicidal ideation (r = 0.22) once age and gender had been controlled for. Conversely, a significant moderate association was found between avoidant attachment security and suicidal ideation (r = 0.36), but the relationship between attachment avoidance and depression was weaker and did not reach significance (r = 0.25). Certainty of mental states, or hypermentalising, was not found to be associated with any of the other psychological variables. A strong association was found between uncertainty of mental states and anxious attachment (r = 0.60), but no association found between uncertainty of mental states and suicidal ideation, when controlling for effects of age and gender.

Guided by Hayes and Rockwood [74], the criteria required for establishing mediation as described by Baron and Kenny [75] was not considered necessary for carrying out mediation analyses. As certainty of mental state, or hypermentalising, was not found to significantly associate with any of the variables of interest at the bivariate level, it was not explored further as a mediating variable. However, although uncertainty of mental state, or hypomentalising, did not correlate at a significant level with suicidal ideation (r = 0.24) or avoidant attachment (r = 0.20) these coefficients suggest a small effect in the hypothesised direction. Therefore, an indirect effect of attachment on suicidal ideation through reflective functioning is plausible through a sequence of steps where attachment affects reflective functioning, which in turn affects suicidal ideation.

When anxious attachment was entered as the independent variable, a significant positive relationship was found with the mediating variable hypomentalisation after adjusting for age, gender and depression symptoms (b = 0.34, *p* < 0.001, BCa CI [0.22, 0.47]. However, there was no significant total, direct or indirect effect of anxious attachment on suicidal ideation (Table 4). When avoidant attachment was entered as the independent variable (Figure 1), no significant coefficients were revealed between avoidant attachment and hypomentalising (path a, *p* = 0.11) or between hypomentalising and suicidal ideation (path b, *p* = 0.11). However, the total effects model was significant, as was the direct effect model once the explanatory mediating variable was added (b = 1.61, *p* = 0.04, BCa CI [0.11, 3.12]). Yet, the absence of a significant indirect effect (b = 0.10, 95% CI [−0.14, 0.84]) confirms that this direct relationship is not mediated by increased hypomentalisation.

Mean comparisons were conducted to compare psychological variables across participants who self-reported a history of suicide attempts never, once or multiple times. Due to a high proportion of the data demonstrating non-normal within-group distribution, the non-parametric Kruskal–Wallis test was run for all analyses. Means, standard deviations, test statistics and pairwise comparisons are presented in Table 5. Significant differences between groups were found for anxious attachment and hypomentalising subscales. For these constructs, pairwise comparisons were carried out to explore where significant differences lie between groups, and calculated effect sizes and adjusted *p*-values are reported in Table 5.

Individuals who reported multiple suicide attempts were found to have greater anxious attachment insecurity compared to those who reported no such history (d = 0.47, adj. *p* < 0.01), but there was no statistical difference between multiple and single attempters (d = 0.11, adj. *p* > 0.99). When comparing single to never attempters, there was a trend towards single attempters being more anxiously attached, but this was not significant when adjustments were made for multiple comparisons (d = 0.37, adj. *p* = 0.08). For avoidant attachment, observing the group means revealed a similar gradual increase in insecurity across the three groups with multiple and never attempts being the most disparate. However, there was no statistically significant difference between the three groups (H = 5.39, *p* = 0.07) and therefore pairwise comparisons were not conducted.

A similar pattern of results was also found for reflective functioning; multiple attempters were significantly more likely to report uncertainty about their own and others’ mental states compared to never attempters (d = −0.38, adj. *p* < 0.01) but not compared to participants with a history of one past attempt (d = −0.11, adj. *p* > 0.99). Likewise, there was no significant difference between single and never attempters once adjustments were made for multiple comparisons (*p* = 0.03, adj. *p* = 0.08). For extreme certainty about mental states, i.e., hypermentalising, no statistical difference or observable trend was found between groups based on their attempt history (H = 1.59, *p* = 0.45).

## 4. Discussion

The aim of the current study was to explore the role of a theoretically determined mediator in the attachment-suicide relationship. More specifically, it was hypothesised that impairments in mentalisation, the ability to understand and interpret actions as expressions of mental states, would bridge the gap between attachment insecurity and suicidal ideation. Overall, the present results did not provide evidence that mentalisation deficits mediate a relationship between attachment insecurity and suicidal ideation. However, several interesting findings emerged that will be discussed in relation to the initial hypotheses.

The main finding of importance was the direct relationship between attachment avoidance and recent suicidal ideation, which remained a moderate effect after controlling for age, gender, and recent symptoms of depression. This finding suggests that being uncomfortable with relational intimacy and overvaluing independence puts individuals at greater risk for experiencing suicidal thoughts. It does, however, contrast with a recent review that found mostly anxious attachment styles to associate with suicidality compared to avoidant attachment [26]. Likewise, preoccupied, fearful or unresolved-disorganised attachment styles, which all reflect a degree of attachment anxiety, have generally been associated with greater suicidal ideation compared to dismissive styles [36,76]. However, a large population-based study demonstrated that scoring high on either dimension increased risk for suicidal ideation [34], and research employing a longitudinal design found only avoidant attachment to predict suicide ideation at 3 month follow up [37].

A case-comparison of adolescent inpatients in the mid-1990s found anxious-related attachment classifications to relate to a history of severe suicidal and/or suicidal behaviour, whereas having a dismissing attachment style was linked to no such history [30]. Adams et al. [30] speculated whether dismissive behaviour such as minimising distress and detaching oneself from attachment-related feelings acts as a protective factor in younger participants but increases risk in the long-term. This may be relevant to the current adult sample; whilst potentially less problematic in adolescence and young adulthood, being high in attachment avoidance may have led to greater social isolation and feelings of loneliness. If individuals high in attachment avoidance are unable to draw on interpersonal resources when they experience acute stress—either due to an absence of meaningful relationships or an unwillingness to trust others—suicide may become a viable option to escape their difficulties.

Overall, the current results do not support the hypothesis that reflective functioning mediates the relationship between attachment security and suicidal ideation. No evidence to support such an effect was available for anxious nor avoidant attachment. Although the relationship between attachment and reflective functioning has been well established in the literature [45,48,50], the relationship between reflective functioning and suicidal ideation had scarcely been examined. Previous literature had relied on proxy measures of reflective functioning or examined attachment and mentalisation in relation to traits of BPD [77]. Therefore, to the authors’ knowledge, this was the first study to directly examine relationships between these three variables using validated measures and may indicate a true non-existent relationship. Furthermore, a multitude of important factors are theorised to mediate the relationship between insecure attachment and suicide. Other psychological constructs outlined in Adams’ original model [25], such as affect regulation, self-esteem or personality traits, may better explain the distal relationship between attachment security and current suicidal thoughts.

The analyses comparing participants based on their self-reported history of suicide attempts also revealed interesting findings. Levels of anxious attachment and hypomentalisation gradually increased across the groups, suggesting that frequency of suicidal behaviour increases as individuals experience greater attachment insecurity, and more impaired mentalisation; group differences which attracted moderate effect sizes. However, no group differences were found between never and single attempters on these measures, which implies there may be characteristic differences between individuals who frequently attempt suicide. For avoidant attachment, although no statistically significant group differences were found, it was interesting to observe that the highest level of avoidant attachment was reported for multiple attempters. One explanation is that individuals with an anxious attachment style have made more frequent past attempts to communicate their pain and seek proximity to others. Comparatively, persons high in attachment avoidance may make less frequent but more lethal attempts, and therefore not report the same self-reported history. This corresponds with research exploring the association between adult attachment and non-suicidal self-injury, were only anxious attachment and fears of abandonment where found to be significant predictors [78].

It should be noted that attempt history was measured using one item from the BSSI, and information was not gathered regarding the nature or lethality of these past attempts. Furthermore, self-report may have been influenced by recall bias; individual’s experiencing current distress may have been more likely to disclose past attempts to express their current pain. In comparison, individuals who were less distressed and can consider their past in hindsight may view the intent behind their actions differently.

Overall, the results indicate a phenomenological distinction between individuals who have attempted suicide, and those who experience suicidal thoughts alone. This is consistent with other diathesis–stress models that view suicide as a continuum from ideation through to attempts and death and have suggested psychological moderators that influence transition through the spectrum [7,13]. Here, higher attachment avoidance was found to associate with current suicidal ideation, whereas higher anxious attachment was found to differentiate individuals with a past suicide attempt, particularly those with multiple attempts.

In his theoretical paper, Adams [25] considered whether different characteristics of internal working models influence the severity of suicidal behaviour and the likelihood of repetition. Adams also differentiated between predominantly interpersonal suicidal actions motivated by an urgent appeal to a threatened attachment relationship, and more despairing and potentially lethal communications driven by strong negative internals models of self and attachment figures. In relation to the current study, participants high in attachment anxiety may have engaged in more frequent suicidal actions to try and promote proximity to their romantic attachment figures. However, these actions may not have been driven by the same degree of suicidal intent felt by avoidant individuals currently experiencing more severe suicidal ideation. In support of this theory, Levi-Belz et al. [42] found suicide attempters high in avoidant attachment to have objectively higher suicidal intent. To confirm whether avoidant attachment predisposes individuals to more fatal outcomes, future research efforts would ideally employ prospective, longitudinal designs and follow individuals over extended time periods and record incidences of high lethality attempts and fatal suicides. As suicide is a rare outcome in the general population, this would likely require a substantial baseline sample.

The recent development of the Reflective Functioning Questionnaire [57] provides researchers with a convenient method to screen individuals for deficits in two broad types of mentalisation. However, it does not claim to capture all dimensions of mentalising or ‘real-time’ mentalising as it unfolds in social interactions. This may have influenced findings as the capacity to mentalise has both trait and state aspects [50]. Although mentalisation difficulties are a trait vulnerability related to disruptions in early attachment [47], the mentalisation-based model argues that impairments are amplified at times of heightened arousal triggered by current stress or interpersonal conflicts [54,79]. In effect, the mentalising system ‘switches off’ when the attachment system is activated. A state–trait interaction is consistent with Adams’ [25] developmental model which places suicide as the consequence of the combination of trait vulnerabilities with current experiences that trigger the attachment system and a period of ‘attachment crisis’.

By limiting assessment to a single time point and using general measures of attachment and mentalisation, the current study was unable to detect these state fluctuations. Whilst the RFQ does attempt to capture how people think and behave when they are feeling angry, insecure or are experiencing strong emotions, if the person is not currently experiencing those difficulties their retrospective recall may not be reliable. Furthermore, the sampling procedure relied upon access to participants who could provide informed consent and tolerate spending time with an unknown researcher completing self-report measures. It is unlikely that nursing staff would have allowed access to participants experiencing current levels of high expressed emotion, and that these individuals could have provided informed consent under those circumstances.

The absence of a mediation effect may also be due to the choice of outcome variable. A significant relationship, albeit with only a weak effect, was found between increased hypomentalisation and high attachment anxiety, yet only attachment avoidance related to increased suicidal ideation. However, group comparisons revealed that a history of multiple suicide attempts was associated with both anxious attachment and hypomentalising; therefore, it would be interesting to explore whether hypomentalising tendencies explain any relationship between anxious attachment and future suicidal behaviour. In contrast, the relationship between avoidant attachment and suicide-related outcomes may be mediated by other psychological constructs. For example, greater loneliness and reduced self-disclosure have been shown to mediate the relationship between attachment avoidance and lethality of suicide attempts [42].

Attachment was measured using the ECR-R; a self-report measure of two relatively orthogonal adult attachment dimensions [65]. This questionnaire has been widely used in psychological research to capture attachment anxiety and avoidance [80]. However, whilst these dimensions represent insecure attachment orientations, they are both organised patterns of relating that enable adults to select strategies that are most adaptive within their relationships [81]. Disorganised attachment, the confused, undirected and inconsistent behaviour often exhibited by infants who view their caregiver as a source of danger and a source of protection, is also a distinct element of the adult attachment system where the central characteristic is a general fear of romantic attachment figures [82]. Research has shown that whilst correlated, disorganised attachment is different from both attachment anxiety and avoidance and persons who are disorganised tend to use conflicting approach and avoidance strategies in their interactions with romantic partners [81]. Previous research that has found unresolved-disorganised to be the predominant attachment pattern in suicidal participants [30]; indicating that this may be an important attachment element that is overlooked in the majority of self-report research. Paetzold et al. [81] have recently developed and validated a dimensional measure for assessing disorganisation in adults that could be administered alongside traditional measures in research that is unable to use extensive interview methods.

There are further limitations that need to be acknowledged when interpreting the findings. Foremost, like most literature in this field the current study employed an observational, cross-sectional design that relied on self-report data. Therefore, causal inferences cannot be drawn from any of the results, and those pertaining to past behaviour (i.e., suicide attempts) may be subject to recall bias. Convenience sampling was used for ease of data collection and to maximise the sample size, which is subject to sampling bias and limits the generalisability of the results as those who volunteered to participate are unlikely to be representative of the wider population. However, the sample did include students, outpatients and persons admitted to inpatient psychiatric hospitals, meaning individuals with a wide range of experiences and severity of suicidal thinking were sampled.

Furthermore, measuring trait differences in attachment and mentalisation conflicts with the theoretical argument that fluctuations under acute stress are key for predicting suicidal behaviour. This raises an important question of how researchers can ethically capture state-level psychological constructs that are activated in times of acute distress. To advance this field, research could consider adopting more intensive, micro-longitudinal designs, such as Ecological Momentary Assessment (EMA) [83], that support real-time assessment at multiple time points. However, even more ecologically valid methodologies would rely on continued compliance from participants during states of high emotional arousal. Experimental studies could be a potential avenue to confirm some of the theorised mechanisms. For example, whether higher-order cognitive functions such as mentalising are compromised during times of acute stress and whether the extent of inhibited mentalisation would vary based on participants’ history of suicidal behaviour and/or degree of attachment insecurity. However, it is debatable whether traditional acute laboratory stressors (e.g., Trier Social Stress Test [84]) could accurately mimic the kind of interpersonal experiences known to trigger crises in attachment.

The Adult Attachment Interview remains the gold standard attachment assessment instrument, and it has the added advantage of being able to capture disorganised attachment which is overlooked in many self-report measures [80]. However, its application is often unfeasible due to the resources and training required to administer the interview. The Experiences in Close Relationships Questionnaire is a sound alternative that is widely used within psychological research and has excellent psychometric properties [65,80]. Furthermore, it captures dimensions of attachment that can detect more subtle differences than traditional classification methods. In the future, it would be advantageous to combine the ECR-R with a dimensional measure of disorganised attachment to capture both organised and disorganised insecure attachment orientations.

As already alluded to, the current study focused on recent suicidal ideation and did not gather more detailed information on participants’ history of suicidal thoughts, communications and attempts over their lifespan. Furthermore, the current findings cannot be generalised to deaths from suicide.

Finally, the current sample size was modest and therefore this study may have been underpowered to detect true findings. The number of variables included in the analyses was restricted due to the small sample size, and more advanced statistical methods that rely on larger sample sizes (i.e., structured equational modelling) were not employed. However, the fact that some significant effects were found despite the small sample is promising, and these may be amplified in future research that is more adequately powered.

## 5. Conclusions

The current research aimed to provide insight into the relationships between attachment, mentalising impairments and suicidality. The present results do not support mentalisation, as measured by the RFQ, as a mediator in the relationship between insecure attachment dimensions and suicidal ideation. Further research is required to confirm whether state variations in mentalisation during periods of attachment crisis could underpin suicidal thoughts and actions. Until this has been achieved, it is difficult to comment on the potential clinical utility of mentalisation-based therapies in relation to reducing risk.

However, the current findings provide further evidence of an association between avoidant attachment and suicidal ideation and highlight ways in which insecure attachment patterns may infer different degrees of suicidal risk. Attachment-based interventions that seek to alter key characteristics of attachment avoidance, such as fear of intimacy and reduced self-disclosure [28], and promote appropriate support seeking in times of distress may help reduce suicidal ideation and subsequent suicide attempts.

## Figures and Tables

**Figure 1 ijerph-18-03090-f001:**
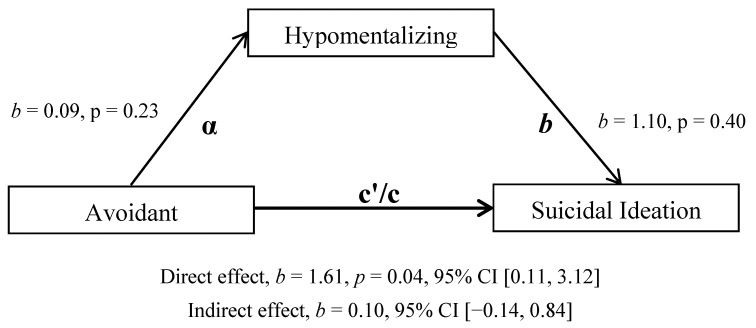
Mediation model with avoidant attachment as the independent variable, hypomentalising as mediator and suicidal ideation score as dependent variable. Controlling for age, gender and current depression.

**Table 1 ijerph-18-03090-t001:** Sociodemographic characteristics.

	Total *N* = 65
Sociodemographic Variables	*N* (%)
**Gender**	
Male	20 (30.8)
Female	45 (69.2)
**Ethnicity**	
White British	54 (83.1)
White Other	4 (6.2)
Other	7 (10.8)
**Educational Attainment, highest level**	
None	4 (6.2)
GCSEs or equivalent	12 (18.5)
A Levels or equivalent	26 (40.0)
Undergraduate degree	9 (13.8)
Postgraduate degree	6 (12.3)
Other	8 (12.3)
**Current Relationship Status**	
Single	44 (67.7)
In a relationship	8 (12.3)
Cohabiting	5 (7.7)
Married	8 (12.3)
**Employment Status**	
Unemployed	9 (13.8)
Unable to work (due to disability, mental health, sickness)	19 (29.2)
Employed	17 (26.2)
Student	19 (29.2)
Retired	1 (1.5)
**Self-reported Psychiatric Diagnosis ^a^**	
None/Not stated	11 (16.9)
Anxiety Disorder	24 (39.6)
Mood Disorder	37 (56.9)
Personality Disorder	14 (21.5)
Psychotic Disorder	10 (15.4)
Other	4 (6.2)
**Self-reported Disability ^a^**	
None	41 (63.1)
Physical	10 (15.4)
Learning Disability/Difficulty	4 (6.2)
Autistic Spectrum Disorder	4 (6.2)
Mental Health	11 (16.9)
Other	1 (1.5)

^a^ Participants could report more than one diagnosis/disability, and therefore the total % may exceed 100.

**Table 2 ijerph-18-03090-t002:** Questionnaire measures: descriptive statistics, normality and reliability tests.

			Kolmogorov–Smirnov Test		Cronbach’s Alpha
	Mean (SD)	Range	Statistic	Sig. Level	α
Suicidal Ideation	14.23 (9.87)	0–35.00	0.13	0.01	0.94
Depression	15.87 (7.38)	0–27.00	0.13	0.01	0.89
Hopelessness	12.09 (6.49)	0–20.29	0.15	0.01	0.94
Attachment Security:					
Anxious Attachment ^a^	4.21 (1.42)	1.06–6.61	0.08	0.20	0.93
Avoidant Attachment ^a^	3.80 (1.44)	1.11–6.67	0.09	0.20	0.94
Reflective Functioning:					
Certainty ^a^	0.66 (0.74)	0–2.83	0.19	<0.001	0.78
Uncertainty ^a^	1.36 (0.88)	0–3.00	0.11	0.04	0.80

^a^*n* = 64 as one participant chose not to complete the ECR-R and RFQ, but their data were retained for those questionnaires completed.

**Table 3 ijerph-18-03090-t003:** Partial correlations, controlling for age and gender.

	Hopelessness	Depression ^a^	Anxious Attachment	Avoidant Attachment	Certainty of Mental State/Hypermentalising	Uncertainty of Mental State/Hypomentalising
Suicidal Ideation	**0.71 **** **[0.59, 0.82]**	**0.48 **** **[0.31, 0.63]**	0.22 [−0.08, 0.50]	**0.36 **** **[0.13, 0.58]**	−0.09 [−0.36, 0.17]	0.24 [−0.04, 0.48]
Hopelessness		**0.60 **** **[0.41, 0.74]**	**0.27 *** **[0.03, 0.49]**	**0.32 *** **[0.11, 0.53]**	−0.12 [−0.39, 0.14]	0.23 [−0.03, 0.45]
Depression ^a^			**0.26 *** **[0.03, 0.45]**	0.25 [−0.01, 0.47]	−0.13 [−0.35, 0.10]	0.23 [−0.01, 0.45]
Anxious Attachment				**0.36 **** **[0.12, 0.56]**	−0.22 [−0.47, 0.02]	**0.60 *** **[0.40, 0.75]**
Avoidant Attachment					0.09 [−0.18, 0.30]	0.20 [−0.07, 0.45]
Certainty of Mental State/Hypermentalising						**−0.61 **** **[−0.74, −0.46]**

BCa bootstrap 95% CIs reported in brackets; * *p* < 0.05, ** *p* < 0.01. ^a^ Partial correlation analyses were repeated following the removal of item nine from the depression scale (PHQ-9) which enquires about recent thoughts of self-harm. This was not found to impact the results.

**Table 4 ijerph-18-03090-t004:** Mediation of attachment security effects on suicidal ideation via hypomentalising, controlling for age and gender.

Independent Variable (Attachment Dimension)	Path a	Path b	Total Effect (c):	Direct Effect (c’)	Indirect Effect	Sobel Test: z-Score (*p* Value)
Anxious	**0.36 [0.23, 0.48]**	1.87 [−0.1.71, 5.46]	1.51 [−0.20, 3.22]	0.84 [−1.30, 2.98]	0.69 [−0.64, 1.93]	1.01 (0.31)
(*controlling for depression*) ^a^	**0.34 [0.22, 0.47]**	1.26 [−2.00, 4.52]	0.72 [−0.87, 2.32]	0.29 [−1.67, 2.25]	0.43 [−0.99, 1.50]	0.75 (0.45)
Avoidant	0.12 [−0.03, 0.26]	1.95 [−0.83, 4.73]	**2.39 [0.81, 3.96]**	**2.16 [0.56, 3.75]**	0.23 [−0.08, 1.02]	0.96 (0.34)
(*controlling for depression*) ^a^	0.09 [−0.06, 0.24]	1.10 [−1.50, 3.71]	**1.71 [0.23, 3.19]**	**1.61 [0.11, 3.12]**	0.10 [−0.14, 0.84]	0.57 (0.57)

BCa bootstrapped 95% CIs reported in brackets. ^a^ Mediation Analyses were also repeated following the removal of item nine from the depression scale (PHQ−9) which enquires about recent thoughts of self-harm. This was not found to impact the results.

**Table 5 ijerph-18-03090-t005:** Differences between participants with no previous suicide attempts, one previous attempt or multiple previous attempts.

	Descriptive Statistics			Kruskal–Wallis Test			Pairwise Comparisons		
	Never (*n* = 17)	Once (*n* = 19)	Multiple (*n* = 29)	H Statistic	Sig.	Never: Once		Never: Multiple	Once: Multiple
Attachment ^a^									
Anxious	3.32 (1.16)	4.38 (1.18)	4.64 (1.50)	10.37	0.01	−0.37		−0.47 **	−0.11
Avoidant	3.03 (1.02)	3.71 (1.51)	4.23 (1.42)	5.39	0.07	-		-	-
Reflective Functioning ^a^									
Certainty	0.59 (0.49)	0.69 (0.73)	0.47 (0.62)	1.59	0.45	-		-	-
Uncertainty	0.82 (0.49)	1.34 (0.89)	1.63 (0.91)	6.431	0.04	−0.22		−0.38 *	−0.16

^a^*n* = 28 for multiple attempt group due to missing questionnaire data; * *p* < 0.05, ** *p* < 0.01.

## Data Availability

The data presented in this study are available on request from the corresponding author. The data are not publicly available due to sensitive information.

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
