# Peer review of "Attachment Security and Suicide Ideation and Behaviour: The Mediating Role of Reflective Functioning"

_ijerph, 2021, doi:10.3390/ijerph18063090_

Round 1

Reviewer 1 Report

ABSTRACT

The reason for studying the relationship between two variables is because it has never been done before. That is not a good enough reason. What is required is a review of the literature leading to a clear research question. I don’t see that Research Question. Be specific about what you expect, so a hypothesis can also be not supported. Mere exploring a topic is not sufficient.

It is not clear to the reader why this is an important topic and why they should read the paper.

Many readers may not know what suicide ideation is, so this term needs to be explained.

INTRODUCTION

The authors supply a detailed review of the literature.

The authors start by giving a review of theories of the causes of suicide. The reader wants to know: how well are the various theories supported? For instance, using Lakatosian research programs: which research programs are progressive and which research programs are degenerating. Surely not all hypotheses are supported equally.

Clement C. Zai, Vincenzo de Luca, John Strauss, Ryan P. Tong, Isaac Sakinofsky, and James L. Kennedy (2012) review the heritability literature of suicide and report a strong influence of genes. Incorporate the findings from behavioral genetics in your paper.

The authors come up with hypotheses. Good. However, the hypotheses need to be made more specific. A positive correlation is expected. So, r = .001 will suffice, and r = .999 as well. Be specific about the effect sizes you expect, and don’t rely solely on significance testing.

Read and cite a recent influential paper by Funder: Funder, D. C., & Ozer, D. J. (2019). Evaluating effect sizes in psychological research: Sense and nonsense. Advances in Methods and Practices in Psychological Science, 2, 156-168.

MATERIAL AND METHODS

Very nice dataset. Sufficient sample size.

A weak point of this study: Only self-report measures are being used. Please note the downsides of the use of only self-report measures instead of higher-quality measures.

Questionnaires have acceptable reliabilities; Certainty of Mental State a bit on the low side.

RESULTS

It is competently analyzed.

The authors work with effect sizes, but there is too much emphasis on outdated significance testing. The American Psychological Association Publication Manual is quite clear that effect sizes are more critical than significance levels.

DISCUSSION

Good, detailed discussion, but too much emphasis on significance testing when drawing conclusions.

Author Response

RESPONSE TO REVIEWERS

REVIEWER 1

  1. ABSTRACT

1.1. The reason for studying the relationship between two variables is because it has never been done before. That is not a good enough reason. What is required is a review of the literature leading to a clear research question. I don’t see that Research Question. Be specific about what you expect, so a hypothesis can also be not supported. Mere exploring a topic is not sufficient.

The rationale for exploring the role of reflective functioning in relation to attachment security and suicidal ideation is, in our opinion, clearly outlined in the introduction. The abstract has now been updated to reflect this and we have explicitly outlined our primary hypothesis under the ‘aims’ subheading (edits highlighted in yellow):

Attachment processes are closely linked to the development of mentalization capabilities, or Reflective Functioning; the ability to understand and interpret self and other behaviour as an expression of mental states. Interventions designed to improve mentalization have been associated with a reduction in suicidal behaviour, yet reflective functioning has rarely been investigated in relation to suicidal ideation and behaviour.

Aim: Further verify the link between adult attachment security and suicidal ideation and examine the hypothesis that deficits in reflective functioning mediate this relationship.

1.2. It is not clear to the reader why this is an important topic and why they should read the paper.

To further clarify why this is an important topic, in the second line of the abstract it is highlighted how suicide is a leading cause of death worldwide. Under the Conclusions subheading, we further reference how increased knowledge can have implications for tailoring interventions aimed at reducing suicide.

1.3. Many readers may not know what suicide ideation is, so this term needs to be explained.

We have amended the wording of the Background section within the abstract to introduce the term ‘suicide ideation’ term to the less familiar reader. The opening line of the Background section now reads:

To understand why attachment difficulties predispose individuals to suicidal thinking (suicide ideation) and behaviour, a leading cause of death, we need to explore the role of pertinent psychological mechanisms.

  1. INTRODUCTION

2.1. The authors start by giving a review of theories of the causes of suicide. The reader wants to know: how well are the various theories supported? For instance, using Lakatosian research programs: which research programs are progressive and which research programs are degenerating. Surely not all hypotheses are supported equally.

The second paragraph of the Introduction has now been revised to better elucidate the level of support currently offered to previous and current theories of the causes of suicide summarised. The following text is now offered:

Although the causes of suicide are not fully understood, it is generally accepted that suicidal ideation and behaviours result from the complex interplay of many factors (O'Connor & Nock, 2014). Psychological explanations have been developed to improve our understanding of suicide Initial explanations emphasised the central role of social connectedness with a lack of social integration proposed to increase the likelihood of suicide (Durkheim, 1897). More contemporary models propose a diathesis-stress framework with a central focus upon cognitive factors (e.g. Baumeister, 1990; Johnson, Gooding, & Tarrier, 2008; O’Connor, 2011; Schotte & Clum, 1987; Williams, 1997, 2001). Most recent suicide theories have been set the challenge of differentiating between individuals who engage in suicidal ideation alone from those whom also engage in suicidal behaviours. The distinction proposed by this ideation to action framework has proven to be especially significant given that most people who think about suicide do not engage in suicide behaviour (Klonsky & May, 2014). Importantly, the Interpersonal-Psychological Theory of Suicidal Behaviour (IPT; Joiner, 2005; Van Orden et al., 2010) presents a desire-capability framework that explained how a desire for suicide could emerge from a sense of social alienation (low belongingness) and burdensomeness but this desire would only be realized (progression from ideation to behaviour) in the presence of a capability to act on such desires.  As such, current understandings of suicide ought to explain not only the initial development of suicidal ideation but also how ideators can be distinguished from suicide attempters. However, theorists have tended to overlook developmental perspectives, for example, how attachment security may contribute to disruptions in relationships and lead to the emergence of suicide ideation and/or suicide behaviour.

2.2. Clement C. Zai, Vincenzo de Luca, John Strauss, Ryan P. Tong, Isaac Sakinofsky, and James L. Kennedy (2012) review the heritability literature of suicide and report a strong influence of genes. Incorporate the findings from behavioral genetics in your paper.

Whilst we do not dispute the potential role of behavioural genetics in the causes of suicide, as well as many other biological markers and potential determinants, the focus of the current paper is specifically upon the potential roles of psychological constructs, namely attachment security and reflective functioning, and so the focus of the current paper remains upon the psychology of suicide.

2.3. The authors come up with hypotheses. Good. However, the hypotheses need to be made more specific. A positive correlation is expected. So, r = .001 will suffice, and r = .999 as well. Be specific about the effect sizes you expect, and don’t rely solely on significance testing. Read and cite a recent influential paper by Funder: Funder, D. C., & Ozer, D. J. (2019). Evaluating effect sizes in psychological research: Sense and nonsense. Advances in Methods and Practices in Psychological Science, 2, 156-168.

We thank the reviewer for highlighting the useful Funder and Ozer (2019) paper for our attention. We agree that presence (or absence) of statistical significance is far less meaningful to interpret compared to the size of the effect. As such, we have revised our wording of Hypothesis one to indicate that a moderate to large effect size is expected when observing the association between anxious and avoidant attachment with suicidal ideation.

  1. MATERIAL AND METHODS

3.1. Self-report measures are being used. Please note the downsides of the use of only self-report measures instead of higher-quality measures.

The limitations of relying on self-report measures are reflected on during the discussion. For example, it is acknowledged that self-report measures are subject to recall bias (line 569), and such measures can only assess trait differences in psychological constructs rather than state fluctuations (lines 662-677). Furthermore, we reference the gold-standard adult attachment interview and acknowledge that by relying on a more feasibly administered self-report questionnaires we have been unable to capture disorganised attachment. Alternative, more ecologically valid methodologies are also suggested (i.e., ecological momentary assessment).

3.2. Questionnaires have acceptable reliabilities, Certainty of Mental State a bit on the low side.

We acknowledge Reviewer 1’s observation that the internal consistency of the RFQ-C is acceptable (α < .80), whereas the other six scales / subscales have good or excellent internal consistency. However, it is encouraging that the internal consistency for the RFQ-C in the current sample (α = .78) was found to be higher than the validity reported in the paper describing the development of this measure, i.e. Fonagy et al., 2016 (a = 0.73). We have added a line to the method section (statistical analysis paragraph) to highlight to the reader that the internal consistency statistics reported for the current sample are comparable to those found in previous research (detailed in the method section, measures paragraph).

Total scales and subscales had acceptable-to-excellent internal consistency (α = 0.78-0.94; see table 2), with Cronbach alpha statistics being comparable to those found in previous research studies.

  1. RESULTS

    4.1. The authors work with effect sizes, but there is too much emphasis on outdated significance testing. The American Psychological Association Publication Manual is quite clear that effect sizes are more critical than significance levels.

    The reporting of the results, and particularly the reporting of the associations presented in Table 3 has now been substantially amened in light of the above comments. The emphasis on significance testing has now been removed with much more importance now attached to the effect sizes of any associations observed.

  1. DISCUSSION

    5.1. Too much emphasis on significance testing when drawing conclusions.

In keeping with our previous response to comment 4.1, the text within the discussion has now been revised such that the previous emphasis placed upon significance testing has now been removed from the revised manuscript.

Reviewer 2 Report

This manuscript explores the relationship between attachment security and suicidal ideation and behaviors, and the potential mediating role of reflective functioning. A review of current literature suggests insecure attachments compromises reflective functioning capability, increases vulnerability to interpersonal threats, thereby contribute to risks for suicidal ideation and behaviors. The authors were the first to investigate these variables directly through mediation analyses. Results did not support a mediating role of reflective functioning, though suggested significant relationship between avoidance attachment and suicidal ideation, as well as different insecure attachments reflect different suicidal risks. The manuscript is well-written, the introduction and discussion are well developed. There are, however, several minor issues that should be addressed before I can recommend this manuscript for publication.

Minor Issues

  • The concepts of hypomentalization and hypermentalization were first introduced in the procedures section, under the Reflective Functioning Questionnaire, each with brief descriptions and few sample items. However, they were repeatedly referenced in the rest of the paper. I believe it is helpful to define and describe these terms in more details in the introduction for better comprehension of the paper.
  • Disorganized attachment was put forward in the introduction but is neither a focus of nor adds incremental value to the aims of the study. I believe it can just be brought up in the discussion section as a limitation and future direction.
  • I would suggest including rationale for inclusion and exclusion criteria.
  • The descriptions of measures mostly only provide internal consistency in previous samples. I would suggest the authors provide information on the validity of the measures in previous samples, as well as the internal consistency in the current sample.
  • Given a small sample size, what was the statistical power, and how did you address it?
  • In reporting statistical outcomes, some places report up to three decimal places (e.g., table three) whereas others report two decimal places (e.g., table four). Also, p-value is reported along with correlation coefficient (e.g., lines 395-397) and is missing for others (e.g., lines 388-391). I would suggest more consistent report of statistics.
  • In reference to lines 566-571, I suggest the authors provide some more practical recommendations on how avoidant attachment predisposes individual to more fatal outcomes can be studied, instead of concluding it is infeasible.
  • Typos: line 155, “metalize”; line 539, “where”.

Author Response

RESPONSE TO REVIEWERS

REVIEWER 2

  1. INTRODUCTION

    1.1. The concepts of hypomentalization and hypermentalization were first introduced in the procedures section, under the Reflective Functioning Questionnaire, each with brief descriptions and few sample items. However, they were repeatedly referenced in the rest of the paper. I believe it is helpful to define and describe these terms in more details in the introduction for better comprehension of the paper.

Hypo- and hyper-mentalisation are now introduced in the introduction where the concept of reflective functioning is first reference. The following text is now offered:

Mentalization, or reflective functioning (used synonymously), refers to the human capacity to understand and interpret one’s own behaviour, and the behaviour of others, as expressions of mental states such as thoughts, feelings, beliefs and desires (Fonagy et al., 2002). Having the ability to form relatively accurate models of the mind, whilst acknowledging the opaqueness of mental states, helps individuals understand and anticipate one another’s actions (Fonagy et al., 2016; Fonagy & Target, 1997). Fonagy and colleagues further outline two subtypes of mentalization impairment: hypomentalization, the extreme difficulty developing complex models of the mind of oneself and others and hypermentalization, the opposite tendency to develop very complex models that have little or no correspondence to observable evidence. Genuine mentalization is a vital skill that allows people to successfully navigate their social world and regulate their affect (Fonagy et al., 2002), and both hypo- and hyper-mentalizing have been implicated in a wide range of psychological disorders (Katznelson, 2014).

1.2. Disorganized attachment was put forward in the introduction but is neither a focus of nor adds incremental value to the aims of the study. I believe it can just be brought up in the discussion section as a limitation and future direction.

The reference to disorganised attachment has been removed from the Introduction and is now only briefly referenced in the Discussion as an area for future research.

  1. METHOD

2.1. I would suggest including rationale for inclusion and exclusion criteria.

The primary inclusion criterion for the current study was self-reported suicidal ideation in the past 12 months. This timeframe has commonly been adopted by other studies within suicide research field (i.e., research undertaken by O’Connor and colleagues), where there needs to be a balance between capturing a group of individual’s where suicidal ideation is an ongoing / proximal experience whilst not making the timeframe too restrictive to hinder recruitment efforts. The rationale has now been included in the method section, inclusion criteria paragraph:

The primary inclusion criterion was self-reported suicidal ideation within the past year. This timeframe is commonly adopted in research examining suicidal ideation (e.g., Branley-Bell, O’Connor, Green, Ferguson, O’Carroll & O’Connor, 2019) to capture a sample where suicidal thinking is a recent or ongoing experience.

2.2. The descriptions of measures mostly only provide internal consistency in previous samples. I would suggest the authors provide information on the validity of the measures in previous samples, as well as the internal consistency in the current sample.

Information on the internal consistency of the measures used in the current sample are provided in Table 2. To draw the reader’s attention to this, table 2 is now referenced in the method section (line 362) as well as the results section.

Total scales and subscales had acceptable-to-excellent internal consistency (α = 0.78-0.94; see table 2), with Cronbach alpha statistics being comparable to those found in previous research studies.

2.3. Given a small sample size, what was the statistical power, and how did you address it?

The authors are in agreement with Reviewer 2 that given the small sample size, the current study may be underpowered. To minimize the impact of a potential lack of power, we limited the number of variables within the statistical analysis. This limitation is acknowledged in the discussion text:

Finally, the current sample size was modest and therefore the study may have been underpowered to detect true findings. The number of variables included in the analyses was restricted due to the small sample size, and more advanced statistical methods that rely on larger sample sizes (i.e., structured equational modelling) were not employed. However, the fact that some significant effects were found despite the small sample is promising, and these may be amplified in future research that is more adequately powered.

  1. RESULTS

3.1. In reporting statistical outcomes, some places report up to three decimal places (e.g., table three) whereas others report two decimal places (e.g., table four). Also, p-value is reported along with correlation coefficient (e.g., lines 395-397) and is missing for others (e.g., lines 388-391). I would suggest more consistent report of statistics.

To improve consistency across the manuscript, we have now removed the p-values from the reporting of correlation coefficients within the main text and provided an indication of statistical significance within Table 3 and 4. We have also now reported all statistical outcomes to 2 decimal places.

  1. DISCUSSION

4.1. In reference to lines 566-571, I suggest the authors provide some more practical recommendations on how avoidant attachment predisposes individual to more fatal outcomes can be studied, instead of concluding it is infeasible.

As suggested, we now offer a more practical recommendation for future research investigating the potential longer-term effects of avoidant attachment difficulties upon suicidal outcomes. The following text has been added (lines 592-596):

To confirm whether avoidant attachment predisposes individuals to more fatal outcomes, future research efforts would ideally employ prospective, longitudinal designs and follow individuals over extended time periods and record incidences of high lethality attempts and fatal suicides. As suicide is a rare outcome in the general population this would likely require a substantial baseline sample.

  1. GENERAL

5.1. Typos: line 155, “metalize”; line 539, “where”.

These typos have been corrected.

Reviewer 3 Report

Dear authors

Thankyou for this contribution to the field of Suicidology. I think it is important, however found it difficult to extract the main findings and structure in the current manuscript. See comments attached.

Thank you for the opportunity to review the manuscript:

 Attachment Security and Suicide Ideation and Behaviour: The Mediating Role of Reflective Functioning

The aim was to understand why attachment difficulties predispose individuals to suicidal ideation and behaviour and explore the role of pertinent psychological mechanisms.

The authors argue that clinicians should focus on reducing attachment anxiety by helping people develop skills in emotional regulation and that interventions aimed at reducing suicidal ideation should focus on reducing attachment avoidance by helping people develop closer relationships with significant others.

The work is therefore relevant for clinicians working with suicidality; however, the aims and results are difficult to follow.

Overall, I think knowledge about these mechanisms in suicidology is of interest to the field, especially since transgenerational suicidality is well documented, and the process and underlying mechanisms are not sufficiently studied.

This study therefore adds novelty and raises important perspectives. However, the main problem is in spite of the fact that the results are interesting, it is difficult to extract them from the current text.

My recommendation is that revisions are necessary in order to highlight the most important findings according to the original aims and try to present it more available for the readers. The current text is difficult to grasp and should be shortened in all sections of the paper.

Please find my comments below.

The abstract should be written in line with the journal requirements and include the subheadings background, method, results and conclusions.

Introduction

The introduction section is too long and needs to be sharpened and focus on the rationale and according aims of the study.

The authors state that theorists have tended to overlook developmental perspectives, for example, how attachment security may contribute to disruptions in relationships and increases in suicide risk. This is an interesting and to my knowledge important perspective although it is not supported with any references.

Methods

 The participants were recruited from both community outpatient and inpatient mental health services, however also via self-recruitment. There is no information about where the final sample was recruited from, that was a convenience sample, the self-selection and generalizability should be described and addressed in the discussion section.

How many filled out the initial screening about recent suicidal ideation?

Is suicidal thoughts last year defined as recent suicidal ideation? Please clarify the terminology.

The self-reported psychiatric diagnoses should be described, and its reliability discussed.

Line 245: …and significant substance use resulting in intoxication at the time of interview.

This is a little confusing, how was this verified, and did they participate in an interview?

The ethics and safety procedures are thoroughly described, however maybe ethics and these measures should be described together in one section? (line 233-235) & (253-256)

Results

Table 2 should be revised, and the numbers aligned.

Table 5

The results are interesting; however, I don’t understand why or how suicide ideation, depression and hopelessness in the three groups with no or previous suicide attempt are relevant for the study aims?

Author Response

RESPONSE TO REVIEWERS

REVIEWER 3

1. Abstract

1.1. The abstract should be written in line with the journal requirements and include the subheadings background, method, results and conclusions.

The abstract has been amended to include subheadings in line with the journal requirements.

  1. Introduction

    2.1. The introduction section is too long and needs to be sharpened and focus on the rationale and according aims of the study.

Following a thorough review by the authors and also in response to reviewers’ comments, a series of edits have now been made to the manuscript which we believe now offers a much sharper focus within the Introduction section, including a clear rationale and set of aims for the study.

2.2. The authors state that theorists have tended to overlook developmental perspectives, for example, how attachment security may contribute to disruptions in relationships and increases in suicide risk. This is an interesting and to my knowledge important perspective although it is not supported with any references.

Within the revised Introduction section, we now include a paragraph that is dedicated to summarising and referencing well-known historical and contemporary psychological models of suicide. This summarises evidence the proposition that none of these models focus upon developmental perspectives, with the exception being Adams’ 1994 model that we reference as an exception. This a novel observation made by the authors in the current paper, and so other references making this point do not exist, to the author’s knowledge.

  1. Methods

    3.1. The participants were recruited from both community outpatient and inpatient mental health services, however also via self-recruitment. There is no information about where the final sample was recruited from, that was a convenience sample, the self-selection and generalizability should be described and addressed in the discussion section.

We are unable to provide further detail on the different referral routes into the study of the various participants, in order to preserve anonymity of all participants. In response to the second point, the use of convenience sampling is now discussed as a potential limitation of the current research, as follows (lines 655-661):

Convenience sampling was used for ease of data collection and to maximise the sample size, which is subject to sampling bias and limits the generalizability of the results as those who volunteered to participate are unlikely to be representative of the wider population. However, the sample did include students, outpatients and persons admitted to inpatient psychiatric hospitals, meaning individuals with a wide range of experiences and severity of suicidal thinking were sampled.

3.2. How many filled out the initial screening about recent suicidal ideation?

In the results section (lines 387/388), it is stated that two participants’ data were excluded because they reported last experiencing suicidal thoughts more than 12 months ago. The primary inclusion criterion was suicidal ideation within the past year, which was outlined in recruitment materials, therefore the screening was just to confirm this. 

3.3. Is suicidal thoughts last year defined as recent suicidal ideation? Please clarify the terminology?

As outlined in our response to Reviewer 2 (comment 2.1), we adopted the timeframe of suicide ideation within the past 12 months, which falls in line with other empirical investigations of recent suicide ideation. In addition to this inclusion criterion, the study administered a questionnaire measure of current suicide ideation (i.e., BSSI) that asked participants to report upon their suicide thoughts and experiences over the past week. To clarify this further, the abstract has been amended to reflect this:

Methods:  Sixty-seven participants who experienced suicidal ideation within the past 12 months completed self-report measures of adult attachment, current suicidal ideation, reflective functioning, depressive symptomology and hopelessness

3.4. The self-reported psychiatric diagnoses should be described, and its reliability discussed.

The authors acknowledge that self-reported diagnoses are not always reliable. However, in the current study diagnoses were only reported for descriptive purposes. The diagnoses are now reported in greater detail in the results section (lines 393-400):

Most participants self-reported at least one psychiatric diagnosis, with Mood Disorders (e.g., Depression, Bipolar), Anxiety Disorders (e.g., Anxiety, PTSD, Social anxiety) and Personality Disorders (e.g., Borderline / Emotionally Unstable Personality Type) most commonly reported. Twenty-four participants also reported having a disability (36.9%), which included physical disabilities (e.g., chronic health conditions, mobility impairments), learning difficulties or disabilities (e.g., Mild Learning Disability, Dyslexia and Dyspraxia), Autistic Spectrum Disorders and Mental Health difficulties (when the participant considered this to be a disability).

3.5. Line 245: …and significant substance use resulting in intoxication at the time of interview. This is a little confusing, how was this verified, and did they participate in an interview?

We are in agreement that the wording of this text was unclear in the original manuscript. To clarify, participants were not screened for substance misuse and excluded on this basis. Rather, if a participant was judged by the researcher to be intoxicated at the time of the interview, then the interview would have been terminated. The inclusion criteria paragraph has been amended accordingly:  

Exclusion criteria included a primary organic mental disorder (e.g., traumatic brain injury, dementia) and judged by the researcher to be intoxicated at the time of interview.

3.6. The ethics and safety procedures are thoroughly described, however maybe ethics and these measures should be described together in one section? (line 233-235) & (253-256)

These sections have now been combined into one section: Sampling and Procedure (lines 246-265).

  1. RESULTS

    4.1. Table 2 should be revised, and the numbers aligned.

The formatting has been reviewed for Table 2, and all numbers are now aligned.

4.2. Table 5 The results are interesting; however, I don’t understand why or how suicide ideation, depression and hopelessness in the three groups with no or previous suicide attempt are relevant for the study aims?

We are in agreement with the above comment. As such, we have removed data on suicide ideation, depression and hopelessness from Table 5 and the related text in the discussion. 

  1. GENERAL

5.1. My recommendation is that revisions are necessary in order to highlight the most important findings according to the original aims and try to present it more available for the readers. The current text is difficult to grasp and should be shortened in all sections of the paper.

In line with all reviewers’ comments, and following a thorough review of the text by the authors, we believe the focus of the paper has been improved such that the important findings of the study are now immediately available to the reader.

Round 2

Reviewer 1 Report

I like to thank the authors for processing the majority of my points. However, some points have not been processed sufficiently, so I simply repeat them.

INTRODUCTION

The authors supply a detailed review of the literature.

The authors start by giving a review of theories of the causes of suicide. The reader wants to know: how well are the various theories supported? For instance, using Lakatosian research programs: which research programs are progressive and which research programs are degenerating. Surely not all hypotheses are supported equally.

Addendum: Make clear statements on how well a theory is supported empirically? Are there meta-analyses to support the claims? Are there convincing arguments? Do outcomes of interview studies all point in the same direction? Have alternative explanations been refuted?

The authors come up with hypotheses. Good. However, the hypotheses need to be made more specific. A positive correlation is expected. So, r = .001 will suffice, and r = .999 as well. Be specific about the effect sizes you expect, and don’t rely solely on significance testing.

Read and cite a recent influential paper by Funder: Funder, D. C., & Ozer, D. J. (2019). Evaluating effect sizes in psychological research: Sense and nonsense. Advances in Methods and Practices in Psychological Science, 2, 156-168.

Addendum. There are only effect sizes for the first hypotheses, not the other ones.

DISCUSSION

Good, detailed discussion, but too much emphasis on significance testing when drawing conclusions.

Addendum: Where are the effect sizes mentioned? The problems with imprecise Research Hypotheses return.

Author Response

REVIEWER 1

  1. INTRODUCTION

  • The authors supply a detailed review of the literature.

The authors start by giving a review of theories of the causes of suicide. The reader wants to know: how well are the various theories supported? For instance, using Lakatosian research programs: which research programs are progressive and which research programs are degenerating. Surely not all hypotheses are supported equally.

Addendum: Make clear statements on how well a theory is supported empirically? Are there meta-analyses to support the claims? Are there convincing arguments? Do outcomes of interview studies all point in the same direction? Have alternative explanations been refuted?

In response to the addendum comments provided by Reviewer 1, we have further expanded our summary of theories of the causes of suicide. Within the amended text below (edits highlighted in yellow), we have now addressed the additional questions asked by Reviewer 1. Specifically, we have highlighted the dominance of Joiner’s IPT theory of suicide, the potential support offered to this theory by the meta-analysis reported by Chu et al (2017) and we have also included the recent criticism of this theory and the associated meta-analysis. We now believe this summary better allows the reader to know how well the currently dominant theory of suicide, namely IPT, is supported.

Lines 50-89:

Although the causes of suicide are not fully understood, it is generally accepted that suicidal ideation and behaviours result from the complex interplay of many factors (O'Connor & Nock, 2014). Psychological explanations have been developed to improve our understanding of suicide. Initial explanations emphasised the central role of social connectedness, with a lack of social integration proposed to increase the likelihood of sui-cide (Durkheim, 1897). More contemporary models propose a diathesis-stress framework with a central focus upon cognitive factors (e.g., Baumeister, 1990; Johnson, Gooding, & Tarrier, 2008; O’Connor, 2011; Schotte & Clum, 1987; Williams, 1997, 2001). Most recent suicide theories have been set the challenge of differentiating between individuals who engage in suicidal ideation alone (ideators) from those whom also engage in suicidal be-haviours (attempters). The distinction proposed by this ideation-to-action framework has proven to be especially significant given that most people who think about suicide do not engage in suicide behaviour (Klonsky & May, 2014).

Importantly, the Interpersonal-Psychological Theory of Suicidal Behaviour (IPT; Join-er, 2005; Van Orden et al., 2010) presented the first desire-capability framework that ex-plained how a desire for suicide could emerge from a sense of social alienation (low be-longingness) and burdensomeness, but this desire would only be realized (progression from ideation to behaviour) in the presence of a capability to act on such desires. In a me-ta-analysis of 122 published and unpublished studies, Chu et al. (2017) examined the decade of research that has subsequently investigated the relationship between IPT con-structs and suicidal ideation and behaviours. Overall, the meta-analysis offered support for IPT, admittedly with weak-to-moderate positive associations reported for thwarted be-longiness (r = 0.37 and r = 0.11) and perceived burdensomeness (r = 0.48 and r = 0.25) with suicidal ideation and suicide attempt history, respectively. Capability for suicide was also significantly associated with suicide ideation and attempts although these effects were weak (rs = 0.09-0.10). Further, in addition to these univariate effects, the interaction of thwarted belongingness and perceived burdensomeness was significantly, but weakly, correlated with suicide ideation (rs = 0.12-0.14). Similarly, the three-way interaction of all IPT constructs was significantly associated with a greater number of suicide attempts, although this interaction effect was also weak (rs = 0.06-0.11). Chu et al (2017) concluded that these findings are largely consistent with the IPT hypotheses, albeit with effect sizes no better that those reported for the “many traditional and often-studied risk factors (e.g., suicide attempt history, demographic variables, psychiatric diagnoses, social factors)” (p.1332). The dominance of Joiner’s IPT (2005) within the suicide literature is not without criticism though. For example, Hjelmeland and Knizek (2020) recently highlighted that the Chu et al (2017) meta-analysis was only able to offer limited support for the IPT and suggested the conclusions to be drawn from this review were “clearly strained”. Indeed, a previous systematic review of studies investigating IPT also suggested that this theory of suicide “may not be as clearly defined nor supported as initially thought” (Ma, Batterham, Calear, & Han, 2016, p. 40).

  • The authors come up with hypotheses. Good. However, the hypotheses need to be made more specific. A positive correlation is expected. So, r = .001 will suffice, and r = .999 as well. Be specific about the effect sizes you expect, and don’t rely solely on significance testing.

Read and cite a recent influential paper by Funder: Funder, D. C., & Ozer, D. J. (2019). Evaluating effect sizes in psychological research: Sense and nonsense. Advances in Methods and Practices in Psychological Science, 2, 156-168.

Addendum. There are only effect sizes for the first hypotheses, not the other ones.

In response to the addendum comment, we have also revised our wording of Hypotheses two and three to indicate that moderate effects are expected for these hypotheses.

Lines 256-261:

H2: The relationship between attachment security and suicidal ideation will be significantly, with a moderate effect, mediated by deficits in reflective functioning.

H3: Participants with a self-reported history of attempted suicide will score significantly higher, with a moderate effect, on measures of attachment security and reflective functioning.

  1. DISCUSSION

  • Good, detailed discussion, but too much emphasis on significance testing when drawing conclusions.

Addendum: Where are the effect sizes mentioned? The problems with imprecise Research Hypotheses return.

Revisions have now been made in several places within the discussion to the wording used to summarise the main findings. The revised wording now more explicitly refers to the effect sizes from the results. Specific changes to wording include:

Lines 527-529:

The main finding of importance was the direct relationship between attachment avoidance and recent suicidal ideation, which remained a moderate effect after controlling for age, gender, and recent symptoms of depression.

Lines 569-572:

Levels of anxious attachment and hypomentalization gradually increased across the groups, suggesting that frequency of suicidal behaviour increases as individuals experience greater attachment insecurity, and more impaired mentalization; group differences which attracted moderate effect sizes.

Lines 639-641:

A significant relationship, albeit with only a weak effect, was found between increased hypomentalization and high attachment-anxiety, yet only attachment avoidance related to increased suicidal ideation.

Reviewer 3 Report

Dear authors,

I have reviewed the new version of the manuscript and think that the changes are in line with all the reviewers comments. The current version is thoroughly revised and more available to readers. Thank you for, to my opinion, an important contribution to the field of Suicidology.  

Author Response

We thank Reviewer 3 for their acceptance of our revised manuscript which has addressed all of their previous comments.